# Hospital Incidence, Sex Disparities, and Perioperative Mortality in Open Surgically Treated Patients with Aneurysms of the Ascending Aorta and Aortic Arch in Switzerland

**DOI:** 10.3390/healthcare12030388

**Published:** 2024-02-02

**Authors:** Anna-Leonie Menges, Alexander Zimmermann, Kerstin Stoklasa, Daniela Reitnauer, Lorenz Meuli, Benedikt Reutersberg

**Affiliations:** Department of Vascular Surgery, University Hospital Zurich, CH-8091 Zurich, Switzerland; anna-leonie.menges@usz.ch (A.-L.M.); kerstin.stoklasa@usz.ch (K.S.); daniela.reitnauer@usz.ch (D.R.); lorenz.meuli@usz.ch (L.M.); benedikt.reutersberg@usz.ch (B.R.)

**Keywords:** thoracic aneurysm, ascending aortic aneurysm, aortic arch aneurysm, acute aortic syndrome, epidemiology, diagnosis-related groups

## Abstract

Objective. To analyze the epidemiological shifts in the incidence of ascending and arch aortic aneurysms (AA) treated with open surgery in the context of evolving endovascular options on a national basis. Methods. Between 1 January 2009 and 31 December 2018, 4388 cases were admitted to the hospital with either ruptured (r)AA or non-ruptured (nr)AA as the primary or secondary diagnosis. Patients were classified as having AA based on inclusion and exclusion criteria. Results. The age-standardized hospital incidence rates for treatment of nrAA were 7.8 (95% confidence interval (CI): 6.9 to 8.7) in 100,000 men and 2.9 (2.4 to 3.4) in 100,000 women and were stable over time. The overall raw in-hospital mortality rate was 2.0% and was significantly lower in males compared to women (1.6% vs. 2.8%, *p* = 0.015). Higher van Walraven scores (OR: 1.08 per point; 95%CI: 1.06 to 1.11; *p* = 0.001) and higher age (OR 1.05 per year; (95%CI: 1.02 to 1.07, *p* = 0.045) were significantly associated with hospital mortality. Conclusions. Endovascular surgery seems to have no influence on hospital incidence in patients treated with conventional surgery for AA in Switzerland. There was a significant reduction in in-hospital mortality in both men and women, with age and the von Walraven score being independent factors for worse outcomes.

## 1. Introduction

Ascending and arch aortic aneurysms (AA) are dilations of the aortic wall that can be catastrophic if not detected and treated in time. Often silent until they reach a significant size or cause complications, these aneurysms can lead to aortic dissection or rupture, both of which are life-threatening [1,2]. The etiology of these aneurysms can be multifactorial, including genetic predisposition, atherosclerosis, hypertension, connective tissue disorders, and other risk factors.

Hospital incidence of AA is a critical measure for healthcare systems, as it provides insight into the prevalence of the disease and the effectiveness of screening programs [3,4]. In recent decades, the broad availability of advanced imaging modalities such as computed tomography (CT) and magnetic resonance imaging (MRI) has revolutionized the early detection and monitoring of these aneurysms, potentially reducing the number of emergency cases [3].

Although population-based studies have been performed for the entire thoracic aorta, there is no reliable data for AA [3,5,6,7]. Research based on the Rochester Epidemiology Project showed an incidence rate of 10.4 per 100,000 person-years for degenerative thoracic aortic aneurysms (TAA) in Olmsted County residents from 1980 to 1994 [2]. However, these data are from patients older than 20 years, and recent epidemiological evidence on these aneurysms is scarce. In addition, no distinction has been made between the ascending aorta/aortic arch and the descending aorta. This is partly because there are only two codes in the 10th revision of the International Classification of Diseases (ICD-10) for documenting TAA. ICD-10 codes I71.1 and I71.2 are used to describe ruptured and non-ruptured aneurysms of the thoracic aorta (rTAA and nrTAA), respectively [8]. Hence, ICD coding does not allow distinction between aneurysms involving only the ascending aorta or aortic arch and those involving only the descending thoracic aorta. This shortcoming hinders data interpretation, as these entities are fundamentally different in their clinical management.

AA are mainly treated with open surgery. Of the various techniques, the best known are the Bentall and the David procedure. In the Bentall procedure, the aortic valve, the aortic root, and the ascending aorta are replaced with a composite graft [9]. In the David procedure, the enlarged aortic root is replaced with a Dacron graft, leaving the natural aortic valve intact [10,11]. In addition to open surgical repair, endovascular treatments are also available in which a stent graft is used to seal the aneurysm and reinforce the aortic wall, also known as thoracic endovascular aortic repair (TEVAR) [7,12]. Endovascular procedures are minimally invasive and are based on adequate sealing in healthy aortic segments. Insufficient sealing can lead to the incomplete exclusion of the aneurysm. Sealing is particularly challenging in the aortic sinus, where the coronary artery originates, and in the aortic arch, where the supra-aortic branches originate. The choice of treatment method depends on several factors, including the patient’s condition, the anatomy of the aneurysm, and the treating physician’s preference. However, AA treatment has evolved over the last decade. With the advancement of endovascular techniques, less invasive approaches, such as endovascular arch repair or endovascular Bentall procedures, have become more common [13,14,15]. The choice between surgical and endovascular intervention usually depends on the anatomical location of the aneurysm, its size, the general health of the patient, and the expertise of the treatment team.

Given these uncertainties, there is a significant gap in our understanding of the epidemiology of ascending aortic arch aneurysms and their perioperative mortality in open surgery. Additionally, this study aimed to analyze epidemiological shifts in the incidence of degenerative AA treated with open surgery in the context of ever-evolving endovascular options.

## 2. Materials and Methods

We performed secondary data analysis using hospital discharge records from the Swiss Federal Statistical Office (SFSO). A detailed methodology for the use of these data has been previously described in a publication on abdominal aortic aneurysm (AAA) [4]. Essentially, every Swiss hospital that treats inpatients is required to report all patient admissions to SFSO annually. This dataset includes various parameters, such as age, sex, up to 50 diagnosis codes, a maximum of 100 procedure codes, details of in-hospital mortality, previous location before admission, type of admission, insurance category, length of time before treatment, total length of hospital stay, length of stay in intensive care, and length of time on mechanical ventilation. Diagnoses were coded according to the International Classification of Diseases 10th revision, while procedures were coded according to the Swiss Classification of Procedures (CHOP). The updated CHOP code list can be found at https://bit.ly/3zYViv6, accessed on 17 January 2024 [16].

It is important to note that, due to privacy standards, patient identifiers were not accessible and institution numbers were coded. This ensures that individual patients undergoing reintervention at different hospital visits remain unidentified.

A review by the local ethics committee (BASEC No Req-2021-01010) confirmed that ethical approval was not required for the analysis of anonymous data. We reported our results according to the STROBE guidelines [17].

### 2.1. Inclusion and Exclusion Criteria

The ICD-10 classification distinguishes thoracic, thoracoabdominal, and abdominal aortic aneurysms. In addition, for each anatomical region, the condition was classified as ruptured or non-ruptured. This means that specificities such as ascending aortic aneurysms, aortic arch aneurysms, and descending aortic aneurysms or their combinations are not explicitly captured in the ICD-10 codes. Therefore, we only included cases with both a relevant ICD-10 diagnosis code and a CHOP procedure code indicating aortic replacement, along with another CHOP code for extracorporeal circulation (ECC). The ICD-10 codes used were I71.1 and I71.2, which cover both rTAA and nrTAA, respectively [8]. The specific CHOP codes used for aortic surgery and ECC are listed in Appendix A [16].

### 2.2. Statistical Analysis

We stratified baseline characteristics and outcomes according to sex. For descriptive statistics, approximately normally distributed continuous variables were reported as the mean and standard deviation (SD). For variables with a skewed distribution, we calculated the median and quartiles (Q1 and Q3). We used the Student’s t-test for normally distributed continuous variables and the Mann–Whitney U test for those with skewed distributions. Categorical variables were described as frequencies and percentages, and comparisons were made using Pearson’s Chi2 test.

We calculated age-standardized cumulative incidences using the 2013 European Standard Population and the Swiss population based on SFSO data for the years 2010 to 2018 [18,19,20]. As the age distribution of the Swiss population in 2009 was not available, we used the age distribution for 2010. The logit-Wald 95% confidence intervals (95%CI) for the direct age-standardized estimates were adjusted as recommended by Altman et al., but on the logit scale [21]. We derived both crude and age-standardized sex-specific hospital mortality rates, adjusting for year of treatment and the sum score of weighted Elixhauser ICD-10 diagnosis groups according to van Walraven’s method [22].

To examine the association between sex and hospital mortality, we constructed a multivariate logistic regression model for the entire cohort. This model was adjusted for potential confounders by including variables such as sex (factor), age (continuous), type of admission (factor), Van Walraven comorbidity score (continuous), insurance class (factor), hospital level (factor), and year of treatment (factor). Regression coefficients are presented as odds ratios (OR) with corresponding 95%CIs.

All analyses were performed using R version 4.2.3 on macOS 12.5.1 [23]. All *p*-values were two-sided, and the significance level was set at 5%.

## 3. Results

Between 1 January 2009, and 31 December 2018, 24,220 patients were admitted to the hospital with either rTAA or nrTAA as the primary or secondary diagnosis. We excluded cases in which nrTAA was the primary or secondary diagnosis but without surgical intervention during hospitalization (n = 18,782). Patients with a concurrent diagnosis of thoracic aortic dissection were also excluded (n = 534). Additionally, cases of rTAA that were transferred (n = 159), those with a secondary diagnosis without treatment (n = 128), and those who survived for more than three days without surgery (n = 209) were excluded. We also excluded cases with duplicate entries (n = 20) from the analysis (Figure 1). This study included 4388 cases, of which 94.1% were treated for nrTAA and 5.9% were treated for rTAA and were rated as AA based on the inclusion and exclusion criteria (see Figure 1).

Table 1 summarizes the baseline characteristics of the study cohort for non-ruptured aneurysms stratified by sex. 73.6% of the patients were male, and the median age was 65 years (Q1, Q3:56–72). Female patients were significantly older than male patients, with a median age of 69 (Q1, Q3:60–75) versus 64 (Q1, Q3:55–71) years (*p* < 0.001). Female patients were significantly more likely to have a diagnosis of chronic obstructive pulmonary disease (COPD) (*p* = 0.018), while men were significantly more frequently affected by coronary heart disease (CHD) (*p* < 0.001). Other comorbidities were similar between men and women.

A similar picture emerged for surgically treated ruptured (r)AA. 31.5% of the treated cases were female and were significantly older than men aged 73 years (Q1, Q3:68, 78) vs. 65 years (Q1, Q3:52, 75) (*p* = 0.012). In terms of comorbidities, the only difference was chronic kidney disease, which was significantly more common in men than in women. (Table 2).

### 3.1. Epidemiology

The age-standardized hospital incidence rates for treatment of non-ruptured (nrAA) were 7.8 (95%CI: 6.9 to 8.7) in 100,000 men and 2.9 (95%CI: 2.4 to 3.4) in 100,000 women. The age-standardized hospital incidence rates for rAA (treated and palliative combined) were 0.4 (95%CI: 0.2 to 0.6) in 100,000 men and 0.3 (95%CI: 0.2 to 0.5) in 100,000 women. The incidence rates were stable in the observed decade for both sexes for nrAAs and rAAs (Figure 2a,b).

### 3.2. Treatment Specifications and Outcomes in nrAA

Table 1 and Table 3 summarize the treatment specifications of the study cohort stratified by sex. The vast majority of the 4127 cases of nrAA were treated in major hospitals. These included university hospitals (55.2%) and large non-university hospitals (30.4%). In nrAA treatment, men received concomitant aortic valve replacement significantly more often than women (16.3% vs. 12.4%, *p* = 0.03). Females received significantly more red blood cells than males (43.2% vs. 27.3%, *p* < 0.001). Females were treated for a significantly longer duration in the ICU than males (median 29 h versus 26 h, *p* = 0.004). The average length of hospital stay was also significantly longer in women (11 days versus 10 days, *p* < 0.001). Men were discharged immediately after hospitalization in 36.2% of cases, compared with 28.7% of women (*p* < 0.001). Only 5.4% (men: 4.9% and women: 7.0%) of all patients were transferred to another acute care hospital after surgical treatment. The data reflect the time to discharge from the hospital where the surgical treatment was performed and therefore do not reflect the total length of stay to discharge from any subsequent inpatient treatment.

The overall raw in-hospital mortality rate for surgically treated nrAAs was 2.0%, which was significantly lower in men than in women (1.6% vs. 2.8%, *p* = 0.015), see Table 3).

The age-adjusted mortality rates for nrAA treatment decreased significantly from 0.9% (95%CI: 0.2 to 4.0%) in 2009 to 0.4% (95%CI: 0.1 to 1.7%) in 2018 in men (*p* = 0.046) and from 1.4% (0.3 to 6.4%) in 2009 to 0.6% (0.1 to 2.8%) in 2018 in women (*p* = 0.045) (Figure 3).

Multivariable logistic regression analysis of hospital mortality for the nrAA cohort showed that sex was not associated with mortality (Figure 4). However, there was a tendency towards higher mortality in females (odds ratio (OR) 1.53; 95% confidence interval CI: 0.95 to 2.43; *p* = 0.077). Higher van Walraven scores (OR: 1.08 per point; 95%CI: 1.06 to 1.11; *p* < 0.001) and higher age (OR 1.05 per year; 95%CI: 1.02 to 1.07; *p* < 0.001) were significantly associated with hospital mortality. Concomitant valve replacement was associated with lower mortality (OR, 0.34; 95%CI: 0.10 to 0.89; *p* = 0.049).

Of note, hospital mortality was lower in major non-university hospitals than in university hospitals and lower in the more recently treated cohort (2014–2018) than in the more historic cohort treated between 2009 and 2013.

### 3.3. Treatment Specifications and Outcomes in rAA

Table 4 summarizes the procedural details and treatment outcomes of all the cases that were surgically treated for rAA. Details of the cases that received palliative care for rAA are summarized in Appendix A. Of note, there were similar tendencies as for the treatment of nrAA, with a higher proportion of no valve treatment, more red blood transfusions, and a longer ICU and hospital stay in females. However, the numbers were much smaller, and none of the differences were statistically significant.

The in-hospital mortality for rTAA decreased by 20.3% in men and 20.6% in women. Owing to the small number of cases, no further statistical analysis was performed.

## 4. Discussion

To our knowledge, this is the first population-based nationwide study of aneurysms in the ascending aorta and the aortic arch. Of the 4388 treated cases in the period analyzed, 94.1% were nrAA and 5.9% were rAA. The hospital incidence of ascending aortic and aortic arch aneurysms was stable for both sexes over the past decade, with 7.8 (95%CI: 6.9 to 8.7) per 100,000 men and 2.9 (2.4 3.4) per 100,000 women for nrAA and 0.4 (95%CI: 0.2 to 0.6) per 100,000 men and 0.3 (0.2 to 0.5) per 100,000 women for rAA. This contrasts with previous studies that have focused on the entire thoracic aorta and showed an increasing incidence [7]. This increase was thought to be mainly due to an increase in CT scans, as these scans often incidentally discover asymptomatic aneurysms. This hypothesis is supported by the significant increase in the number of chest CT scans performed over time [24]. This increased incidence in men is consistent with findings from the Swedish National Health Care Register, showing a higher rate of surgical intervention in men [3]. Nevertheless, the incidence of aortic dissection and aortic rupture remains stable during the same period, ranging from 1.6 to 5 per 100,000 persons/year [24,25,26].

There has been a more pronounced increase in TAA and dissection incidence among women than among men, possibly linked to increased awareness of severe female vascular diseases, which was also not reflected in our data for AA in Switzerland [27].

Sampson et al. reported an increasing global mortality rate for all aortic aneurysms from 1990 to 2010, from 2.49 to 2.78 per 100,000 persons/year. The Rochester Epidemiology Project database showed a 5-year survival rate of only 62.5% for thoracic aortic aneurysms, with age and smoking being associated with all-cause mortality [6].

The overall hospital mortality in surgically treated nrAAs was 2.0% and lower compared to countries such as the United States or the Netherlands [28,29]. Interestingly, the crude mortality was significantly higher in women than in men, which is in line with most analyses of aortic surgery [7,29,30,31,32]. However, a multivariable analysis adjusted for the observed age difference and other parameters showed that mortality did not differ significantly between the sexes. Of note, women were on average 5 years older at the time of surgery, and increasing age was significantly associated with hospital mortality, as previously shown [4,7,33]. The average hospital stay was significantly longer for women, and the rate of direct discharge to home was lower, presumably indicating higher morbidity in females.

The multivariable analysis further revealed that a higher morbidity score (van Walraven score) was associated with higher mortality. On the other hand, a more recent treatment (surgery in the second half of the study period), concomitant aortic valve replacement, and treatment in a major hospital (non-university) were associated with lower mortality. While age and comorbidities are known risk factors, the better outcome in major hospitals compared to university hospitals was surprising (OR 0.50; 95%CI: 0.28 to 0.87; *p* = 0.018). A clear association between treatment volume and outcome has been demonstrated in the field of aortic surgery [34]. In particular, for operations on the ascending aorta, aortic arch for aortic dissection type A, better results were demonstrated with higher individual and institutional case numbers [35,36]. As higher case numbers are assumed at university hospitals, our results contrast these findings. However, this administrative data does not allow adjustments for clinical parameters and thus obscures further analyses. One explanation could be that more training is carried out at university hospitals and there is a greater turnover of staff, which in turn leads to lower individual case numbers. In addition, it can be assumed that doctors in training have lower skill levels, which in turn leads to poorer results [37]. However, this is rather speculative.

The in-hospital mortality rate for operated ruptures in Switzerland was 20.4% and stable in the observed decade. There is only historic data available to put these findings into context. An analysis of the US Medicare population in 2011 showed that the all-cause hospital mortality rate after open surgical treatment for ruptures of thoracic aortic aneurysm was 45% [38]. The substantially better treatment results in Switzerland are likely associated with the high turn-down rate of almost 60%, which induces selection bias and hinders a direct comparison. More historic data from Ontario, Canada, showed an overall mortality rate of 80–100% for rAA between 1998 and 2016 but included patients without surgical treatment [39,40].

Open surgical repair is still considered the gold standard for the treatment of pathologies involving the ascending aorta and aortic arch [41,42]. However, a significant proportion of patients with these pathologies have relevant concomitant cardiopulmonary diseases that preclude these highly invasive procedures. The endovascular revolution began in the late 1980s and has become the gold standard for the treatment of descending aortic pathologies [43]. Likewise, the majority of patients with infrarenal aortic diseases are nowadays treated endovascularly in most Western countries. These successes have led to an extension of the endovascular concept to the more complex thoracoabdominal aortic segment. The first branched thoracic stent graft was performed in 1996 on a patient with an aortic dissection [44]. However, it is only in the last 10–15 years that this technology has become established, standardized, and widespread.

The aortic root and the ascending aorta constitute unique anatomical and physiological characteristics. The rapid blood flow, the high wall tension, the proximity to the aortic valve and coronary ostia, as well as the four-dimensional pulsatile motion during cardiac and respiratory cycles, represent the ultimate challenges to endovascular therapy [45]. Despite these challenges, attempts to treat the ascending aorta endovascularly have been made since 2000 and have gained increasing attention as an alternative treatment strategy, especially for critically ill patients [41]. In recent years, there have been growing case series on the endovascular treatment of non-dissected diseases of the ascending aorta [42]. The currently available prostheses on the market require specific anatomical conditions. Moreover, most are not directly available off-the-shelf but are custom-made for individual patients, posing a challenge in emergency situations due to the time required for production. Nevertheless, viable solutions exist with the appropriate equipment. For instance, in emergency situations where cerebral perfusion is assured (e.g., with veno-arterial ECMO), in situ laser fenestration can be used to address pathologies in the aortic arch [46].

With endovascular aortic arch repair being established in selected patients, endovascular therapy has reached the ascending aorta and, with it, its last bastion. It is particularly interesting for patients with a high surgical risk and could offer these patients potentially life-saving therapy that would otherwise not be surgically treated [13,43]. While the introduction of endovascular therapy in the descending aorta and abdominal aorta has led to a significant reduction in conventional open surgery, this trend has not been demonstrated for the ascending aorta or the aortic arch [4,7]. Likewise, there was no decreasing incidence of open repair of ascending or arch pathologies in Switzerland. It seems like the endovascular therapy in Switzerland is currently complementing open repair and does not lead to a decrease in open repair.

Diseases of the proximal aorta are complex and associated with high procedural risks. This requires tailor-made treatment strategies that are established by a multidisciplinary team, including cardiac surgeons, vascular surgeons, interventional radiologists, cardiologists, and anesthesiologists.

There are several limitations to this analysis, mainly due to the use of administrative rather than clinical data. The inherent strengths and limitations of routinely collected administrative data have been well documented [44].

The major limitation of this study was the lack of a specific ICD code for aneurysms of the ascending aorta, aortic arch, and descending aorta. Instead, the current classification bins these entities into one code for unruptured cases and one code for ruptured cases. This makes it difficult to distinguish between aneurysms involving the ascending aorta and those involving the descending aorta. The combination of ICD and procedure coding allows the identification of open surgical cases of thoracic aortic aneurysms. However, there may be rare cases in which ECC is not coded. As this would have a major impact on reimbursement, this problem is likely to be small. Furthermore, some cases captured in this study may have been treated with ECC for an isolated descending aortic aneurysm. Robust data on the epidemiology of this rare pathology are lacking, but it is likely that most of these cases are treated with endovascular therapy; thus, the number of cases erroneously included in this study may be relatively small. Furthermore, estimating the proportion of palliative treatments for rTAA remains challenging because these cases cannot be distinguished from conservatively treated ruptures of descending thoracic aortic aneurysms.

There are other minor limitations that need to be discussed, such as the potential for bias due to coding errors that cannot be quantified and eliminated. However, for difficult outcomes, such as all-cause in-hospital mortality, the risk is likely to be low. Furthermore, the data do not allow independent observation of patients treated twice for the same ICD code, leading to double counting and potential bias in incidence rates. Although the proportion of affected patients is unknown, it is likely negligible. Furthermore, the comparison of mortality rates with international data is limited because adjustments could only be made for age, sex, and the van Walraven comorbidity index, which does not fully capture the cardiovascular risk profile or functional capacity of treated individuals. Finally, the data do not differentiate between individuals with permanent residences in Switzerland and those with non-permanent residences (e.g., travelers and tourists). As a result, the actual incidence rates in Switzerland may differ slightly from those reported.

Despite these limitations, this analysis has a distinct advantage over registry data in terms of complete coverage of the population. It clearly showed robust epidemiologic trends because the assumptions and limitations remained the same throughout the reporting period.

## 5. Conclusions

This population-based national study of aneurysms of the ascending aorta and aortic arch showed that despite the international introduction of endovascular therapy, there was no reduction in the hospital incidence of patients treated with conventional surgery in Switzerland. At the same time, there was a significant reduction in in-hospital mortality in both men and women, with age and the von Walraven score being independent factors for worse outcomes.

## Figures and Tables

**Figure 1 healthcare-12-00388-f001:**
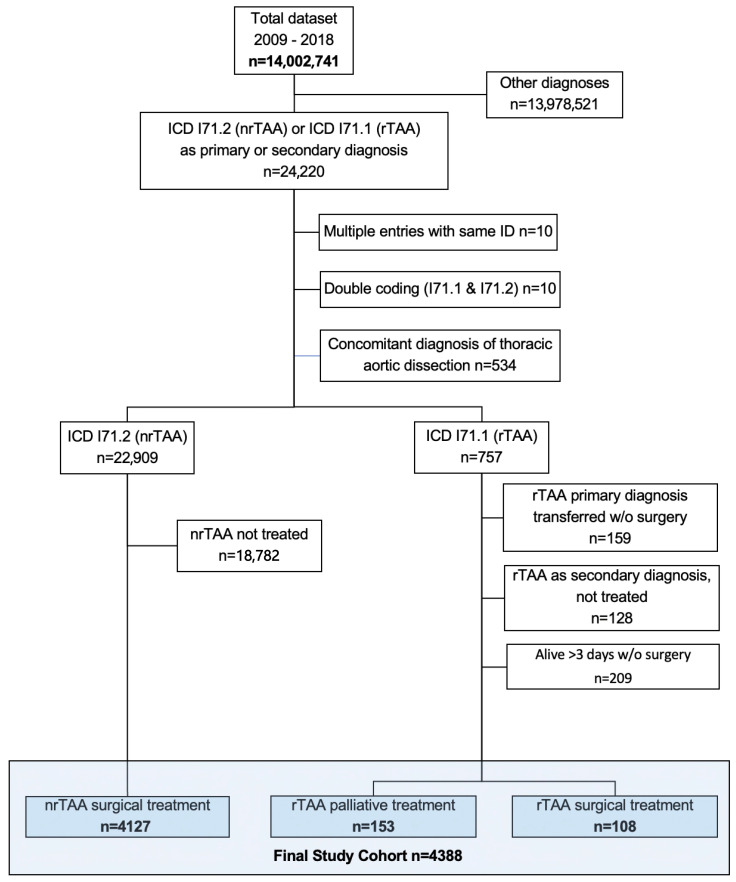
Patient flow. The dataset encompassed all hospital admissions within the Swiss population from the years 2009 to 2018. ICD: International Classification of Diseases (version 10); rTAA: ruptured thoracic aortic aneurysm; nrTAA: non-ruptured thoracic aortic aneurysm.

**Figure 2 healthcare-12-00388-f002:**
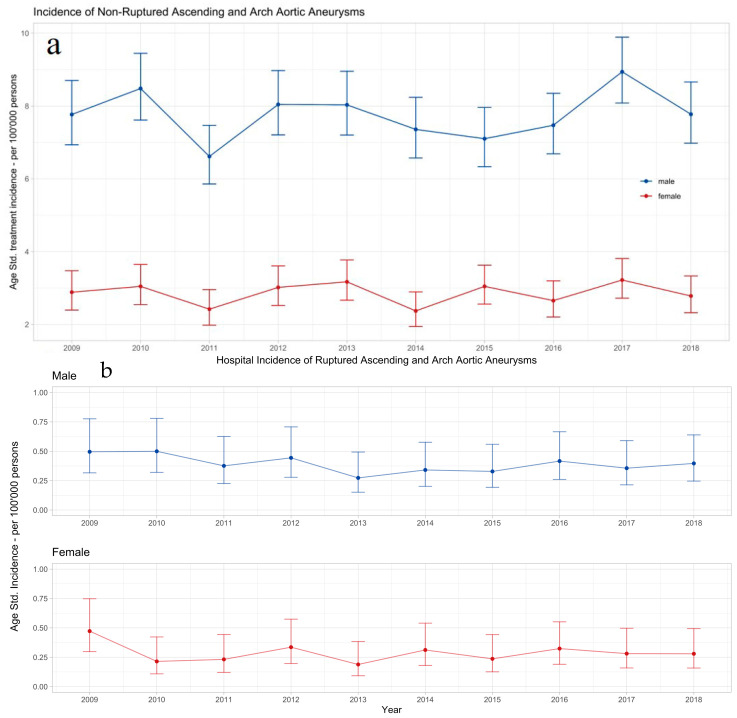
Hospital incidence of non-ruptured ascending and arch aortic aneurysm (**a**) and ruptured ascending and arch aortic aneurysm (**b**). The standardized incidence rates are presented with 95% confidence intervals for each year and stratified by sex.

**Figure 3 healthcare-12-00388-f003:**
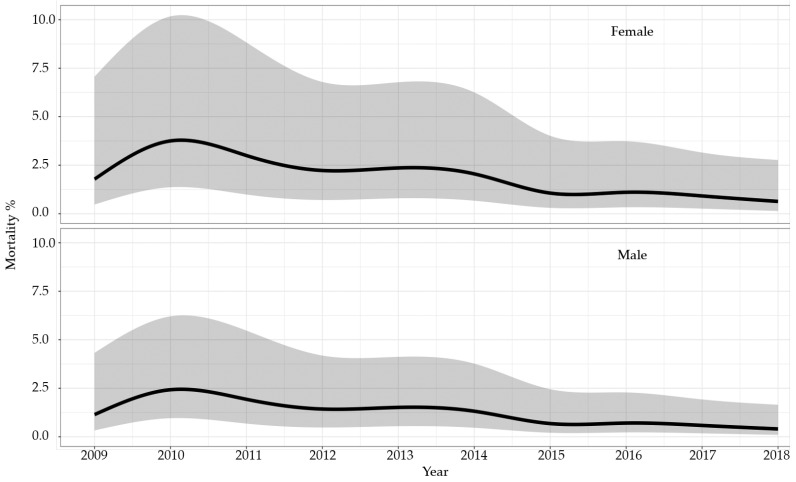
Hospital mortality for non-ruptured ascending and arch aortic aneurysms (nrAA). Smoothed and adjusted hospital mortality of male and female cases in Switzerland between 2009 and 2018. The shadowed area around the curves indicates the 95% confidence interval. Mortality rates were adjusted for age, year of treatment, and van Walraven score.

**Figure 4 healthcare-12-00388-f004:**
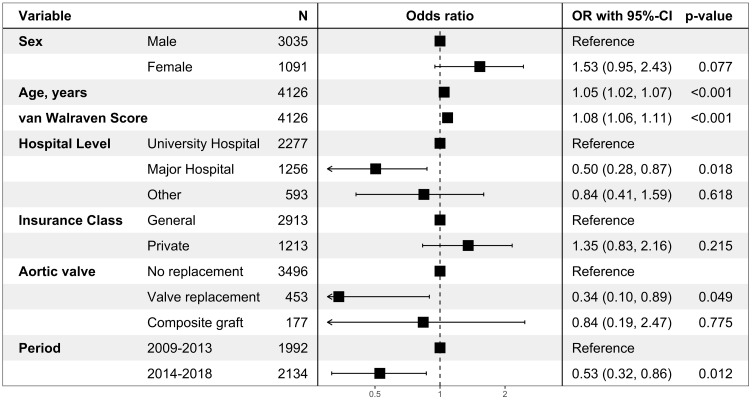
Multivariable regression analysis on hospital mortality. The multivariable logistic regression model for nrTAA (n = 4126). One case was deleted due to missing data. Odds ratios are presented with corresponding 95% confidence intervals and *p*-values.

**Table 1 healthcare-12-00388-t001:** Baseline characteristics of non-ruptured ascending and arch aortic aneurysms (nrAA).

	Male (*N* = 3036)	Female (*N* = 1091)	Total (*N* = 4127)	*p*-Value
**Age, years**	64 (55, 71)	69 (60, 75)	65 (56, 72)	<0.001
**van Walraven score**	7 (3, 14)	8 (2, 14)	7 (3, 14)	0.411
**Coronary artery disease**	1126 (37.1%)	306 (28.0%)	1432 (34.7%)	<0.001
**Chronic heart failure**	519 (17.1%)	186 (17.0%)	705 (17.1%)	0.972
**Cerebrovascular disease**	246 (8.1%)	80 (7.3%)	326 (7.9%)	0.419
**Arterial hypertension**	1455 (47.9%)	543 (49.8%)	1998 (48.4%)	0.295
**COPD**	333 (11.0%)	149 (13.7%)	482 (11.7%)	0.018
**Diabetes mellitus**	240 (7.9%)	67 (6.1%)	307 (7.4%)	0.057
**Chronic kidney disease**	304 (10.0%)	109 (10.0%)	413 (10.0%)	0.983
**Cancer**	18 (0.6%)	3 (0.3%)	21 (0.5%)	0.206
**Obesity**	143 (4.7%)	67 (6.1%)	210 (5.1%)	0.065
**Marfan syndrome**	49 (1.6%)	15 (1.4%)	64 (1.6%)	0.584
**Level of care**				0.372
University Hospital	1690 (55.7%)	587 (53.8%)	2277 (55.2%)	
Major Hospital	922 (30.4%)	334 (30.6%)	1256 (30.4%)	
Other	424 (14.0%)	170 (15.6%)	594 (14.4%)	
**Location before admission**				0.344
Home	2832 (93.3%)	1016 (93.1%)	3848 (93.2%)	
Nursing home	3 (0.1%)	1 (0.1%)	4 (0.1%)	
Other	30 (1.0%)	5 (0.5%)	35 (0.8%)	
Acute care hospital	171 (5.6%)	69 (6.3%)	240 (5.8%)	
**Treatment period**				0.921
2009–2013	1464 (48.2%)	528 (48.4%)	1992 (48.3%)	
2014–2018	1572 (51.8%)	563 (51.6%)	2135 (51.7%)	

Data were complete. The counts are presented as percentages in parentheses. Continuous variables were presented as medians and quartiles (Q1 and Q3). COPD: chronic obstructive pulmonary disease.

**Table 2 healthcare-12-00388-t002:** Baseline characteristics of ruptured ascending and arch aortic aneurysms (surgically treated) (rAA).

	Male (*N* = 74)	Female (*N* = 34)	Total (*N* = 108)	*p*-Value
**Age, years**	65 (52, 75)	73 (68, 78)	69 (54, 75)	0.012
**van Walraven score**	13 (3, 20)	12 (3, 16)	13 (3, 19)	0.540
**Coronary artery disease**	17 (23.0%)	7 (20.6%)	24 (22.2%)	0.782
**Chronic heart failure**	8 (10.8%)	6 (17.6%)	14 (13.0%)	0.326
**Cerebrovascular disease**	14 (18.9%)	6 (17.6%)	20 (18.5%)	0.874
**Arterial hypertension**	30 (40.5%)	17 (50.0%)	47 (43.5%)	0.357
**COPD**	7 (9.5%)	4 (11.8%)	11 (10.2%)	0.713
**Diabetes mellitus**	2 (2.7%)	4 (11.8%)	6 (5.6%)	0.056
**Chronic kidney disease**	9 (12.2%)	0 (0.0%)	9 (8.3%)	0.034
**Cancer**	1 (1.4%)	0 (0.0%)	1 (0.9%)	0.496
**Obesity**	1 (1.4%)	0 (0.0%)	1 (0.9%)	0.496
**Marfan syndrome**	1 (1.4%)	1 (2.9%)	2 (1.9%)	0.569
**Level of care**				0.680
University hospital	50 (67.6%)	22 (64.7%)	72 (66.7%)	
Major hospital	19 (25.7%)	8 (23.5%)	27 (25.0%)	
Other	5 (6.8%)	4 (11.8%)	9 (8.3%)	
**Location before admission**				0.821
Home	42 (56.8%)	19 (55.9%)	61 (56.5%)	
Nursing home	0 (0.0%)	1 (2.9%)	1 (0.9%)	
Acute care hospital	32 (43.2%)	14 (41.2%)	46 (42.6%)	
**Treatment period**				0.420
2009–2013	33 (44.6%)	18 (52.9%)	51 (47.2%)	
2014–2018	41 (55.4%)	16 (47.1%)	57 (52.8%)	

Data were complete. The counts are presented as percentages in parentheses. Continuous variables were presented as medians and quartiles (Q1 and Q3). COPD: chronic obstructive pulmonary disease.

**Table 3 healthcare-12-00388-t003:** Treatment and outcome of non-ruptured ascending and arch aortic aneurysms (nrAA).

	Male (*N* = 3036)	Female (*N* = 1091)	Total (*N* = 4127)	*p*-Value
**Aortic valve management**				0.003
No replacement	2540 (83.7%)	956 (87.6%)	3496 (84.7%)	
Valve replaced, allograft	2 (0.1%)	0 (0.0%)	2 (0.0%)	
Valve replaced, mechanical	58 (1.9%)	15 (1.4%)	73 (1.8%)	
Valve replaced, biological	284 (9.4%)	94 (8.6%)	378 (9.2%)	
Composite graft, biological	116 (3.8%)	23 (2.1%)	139 (3.4%)	
Composite graft, mechanical	36 (1.2%)	3 (0.3%)	39 (0.9%)	
**Hybrid procedure**	19 (0.6%)	11 (1.0%)	30 (0.7%)	0.202
**Length of Ssay ICU, hours**	26 (21, 58)	29 (22, 68)	27 (21, 62)	0.004
**Length of stay (swissdrg), days**	10 (9, 14)	11 (9, 15)	11 (9, 14)	< 0.001
**Packed red blood cells**				< 0.001
0	2208 (72.7%)	620 (56.8%)	2828 (68.5%)	
1–5	639 (21.0%)	383 (35.1%)	1022 (24.8%)	
>5	189 (6.2%)	88 (8.1%)	277 (6.7%)	
**Fresh frozen plasma**				0.052
0	2877 (94.8%)	1034 (94.8%)	3911 (94.8%)	
1–5	107 (3.5%)	48 (4.4%)	155 (3.8%)	
>5	52 (1.7%)	9 (0.8%)	61 (1.5%)	
**Platelet concentrate**				0.050
0	2940 (96.8%)	1072 (98.3%)	4012 (97.2%)	
1–5	91 (3.0%)	18 (1.6%)	109 (2.6%)	
>5	5 (0.2%)	1 (0.1%)	6 (0.1%)	
**Complications**				
Myocardial infarction	46 (1.5%)	15 (1.4%)	61 (1.5%)	0.742
Acute stroke	58 (1.9%)	11 (1.0%)	69 (1.7%)	0.046
Acute paraplegia	3 (0.1%)	2 (0.2%)	5 (0.1%)	0.491
Acute renal failure	2 (0.1%)	2 (0.2%)	4 (0.1%)	0.285
CVVHD	71 (2.3%)	25 (2.3%)	96 (2.3%)	0.929
Acute mesenteric ischemia	6 (0.2%)	4 (0.4%)	10 (0.2%)	0.330
Large bowl resection	4 (0.1%)	1 (0.1%)	5 (0.1%)	0.744
Small bowl resection	3 (0.1%)	2 (0.2%)	5 (0.1%)	0.491
Acute limb ischemia	15 (0.5%)	20 (1.8%)	35 (0.8%)	< 0.001
Crural fasciotomy	3 (0.1%)	2 (0.2%)	5 (0.1%)	0.491
Amputation	1 (0.0%)	0 (0.0%)	1 (0.0%)	0.549
**Destination after discharge**				< 0.001
Home	1098 (36.2%)	313 (28.7%)	1411 (34.2%)	
Nursing home	10 (0.3%)	3 (0.3%)	13 (0.3%)	
Other	18 (0.6%)	5 (0.5%)	23 (0.6%)	
Rehabilitation	1712 (56.4%)	663 (60.8%)	2375 (57.5%)	
Acute care hospital	148 (4.9%)	76 (7.0%)	224 (5.4%)	
Deseased	50 (1.6%)	31 (2.8%)	81 (2.0%)	
**Hospital Mortality**	50 (1.6%)	31 (2.8%)	81 (2.0%)	0.015

Data were complete. Continuous variables were summarized as medians and quartiles (Q1, Q3). ICU, intensive care unit; CVVHD, continuous venovenous hemodialysis.

**Table 4 healthcare-12-00388-t004:** Treatment and outcome of ruptured ascending and arch aortic aneurysms (surgically treated) (rAA).

	Male (*N* = 74)	Female (*N* = 34)	Total (*N* = 108)	*p*-Value
**Aortic valve management**				0.485
No replacement	66 (89.2%)	32 (94.1%)	98 (90.7%)	
Valve replaced, mechanical	1 (1.4%)	0 (0.0%)	1 (0.9%)	
Valve replaced, biological	4 (5.4%)	0 (0.0%)	4 (3.7%)	
Composite graft, biological	2 (2.7%)	2 (5.9%)	4 (3.7%)	
Composite graft, mechanical	1 (1.4%)	0 (0.0%)	1 (0.9%)	
**Hybrid procedure**	1 (1.4%)	0 (0.0%)	1 (0.9%)	0.496
**Length of stay ICU, hours**	55 (20, 132)	58 (20, 142)	58 (20, 137)	0.924
**Length of stay (swissdrg), days**	11 (8, 17)	15 (8, 20)	12 (8, 19)	0.240
**Transfusion Ec**				0.444
0	33 (44.6%)	11 (32.4%)	44 (40.7%)	
1–5	21 (28.4%)	13 (38.2%)	34 (31.5%)	
>5	20 (27.0%)	10 (29.4%)	30 (27.8%)	
**Transfusion FFP**				0.223
0	64 (86.5%)	33 (97.1%)	97 (89.8%)	
1–5	7 (9.5%)	1 (2.9%)	8 (7.4%)	
>5	3 (4.1%)	0 (0.0%)	3 (2.8%)	
**Transfusion Tc**				0.667
0	67 (90.5%)	30 (88.2%)	97 (89.8%)	
1–5	6 (8.1%)	4 (11.8%)	10 (9.3%)	
>5	1 (1.4%)	0 (0.0%)	1 (0.9%)	
**Complications**				
Myocardial infarction	1 (1.4%)	1 (2.9%)	2 (1.9%)	0.569
Acute stroke	1 (1.4%)	1 (2.9%)	2 (1.9%)	0.569
Acute paraplegia	0	0	0	n.a.
Acute renal failure	1 (1.4%)	0 (0.0%)	1 (0.9%)	0.496
CVVHD	2 (2.7%)	2 (5.9%)	4 (3.7%)	0.416
Acute mesenteric ischemia	4 (5.4%)	2 (5.9%)	6 (5.6%)	0.920
Large bowl resection	2 (2.7%)	0 (0.0%)	2 (1.9%)	0.333
Small bowl resection	2 (2.7%)	0 (0.0%)	2 (1.9%)	0.333
Acute limb ischemia	5 (6.8%)	0 (0.0%)	5 (4.6%)	0.121
Crural fasciotomy	0	0	0	n.a.
Amputation	0	0	0	n.a.
**Destination after discharge**				0.205
Home	14 (18.9%)	3 (8.8%)	17 (15.7%)	
Rehabilitation	27 (36.5%)	19 (55.9%)	46 (42.6%)	
Acute care hospital	18 (24.3%)	5 (14.7%)	23 (21.3%)	
Deseased	15 (20.3%)	7 (20.6%)	22 (20.4%)	
**Hospital Mortality**	15 (20.3%)	7 (20.6%)	22 (20.4%)	0.970

Data were complete. Continuous variables were summarized as medians and quartiles (Q1, Q3). ICU, intensive care unit; CVVHD, continuous venovenous hemodialysis.

## Data Availability

The full dataset can be requested from the Federal Statistical Office of Switzerland, Espace de l’Europe 10, and CH-2010 Neuchâtel, Switzerland.

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
