# Peer review of "Hospital Incidence, Sex Disparities, and Perioperative Mortality in Open Surgically Treated Patients with Aneurysms of the Ascending Aorta and Aortic Arch in Switzerland"

_healthcare, 2024, doi:10.3390/healthcare12030388_

Round 1

Reviewer 1 Report

Comments and Suggestions for Authors

The authors conducted a detailed analysis of epidemiological shifts in patients undergoing aortic arch surgery using clinical patient data from the Swiss Federal Statistical Office. This study had a large sample size and clear inclusion and exclusion criteria. However, I believe that the study still needs to be improved in the following points.

Major concern

1. Table1-2 analyses the baseline characteristics of ascending and arch aortic aneurysms. The author trying to analysis of various risk factors and gender differences on the impact of the end. However, these risk factors may already differ according to sex. Even if there is a statistical difference, further analysis of the distribution of different risk factors in the normal population should be conducted. These conclusions the author doesn't seem to be enough creative.

 2. The content of this study needs to be further expanded. Additional analysis of some laboratory test data or imaging data of these patients may have more important significance in guiding clinical diagnosis and treatment.

Author Response

Reviewer 1.

The authors conducted a detailed analysis of epidemiological shifts in patients undergoing aortic arch surgery using clinical patient data from the Swiss Federal Statistical Office. This study had a large sample size and clear inclusion and exclusion criteria. However, I believe that the study still needs to be improved in the following points.

Major concern

  1. Table1-2 analyses the baseline characteristics of ascending and arch aortic aneurysms. The author trying to analysis of various risk factors and gender differences on the impact of the end. However, these risk factors may already differ according to sex. Even if there is a statistical difference, further analysis of the distribution of different risk factors in the normal population should be conducted. These conclusions the author doesn't seem to be enough creative.

Thank you for this excellent remark. This is in fact an interesting consideration. There are several assumptions that must be done when doing a logistic regression analysis like we did. One of them is the assumptions of independence. In this context it means that the effect of age on the outcome (and all the other variables included) is depending on the sex of the patient. This is generally not the case in cardiovascular research, but it is reasonable to challenge this assumption.

An alternative way to analyze the data in a situation where a classic logistic approach would be incorrect is the use of a multilevel multivariable analysis. As suggested by the reviewer we conducted another analysis of the data using a multilevel multivariable logistic regression model and thereby clustered the data by sex (sex = level variable). This allows all the other variables to have different coefficients and the intercept to be different for both sexes.

The results can be seen here:

The model shows almost identical coefficients for all the included variables as the original model:

Thus, we decided to stick to the original model and keep sex as a simple binary variable without multilevel adjustment.

The authors do not fully understand the second part of the comment. What was meant by the reviewer? We do not have data of the entire population describing the distribution of cardiovascular risk factors and if we had so, we do not see a plausible method to include such information into the analysis. If we misunderstood the comment, we ask for a clarification by the reviewer, thank you!

  1. The content of this study needs to be further expanded. Additional analysis of some laboratory test data or imaging data of these patients may have more important significance in guiding clinical diagnosis and treatment.

We thank the reviewer for this recommendation. However, as this is a secondary data analysis, as explained in the methods section, and is based on data that is not primarily intended for research but for billing purposes (DRG system), the data requested by the reviewer is not available to us for analysis. We are therefore unable to carry out these requested analyses.

Reviewer 2 Report

Comments and Suggestions for Authors

Congratulations for your work. Although this study is an retrospective study from a national databasis, still it is quite interesting. You mention in your manuscript that similar studies from other national databases have already been published (eg. the Swedish one). However, I believe that your study adds knowledge to the existing. It is a well designed and presented study. I would omit some repetitions in the discussion. For example, you mention the advances of endovascular procedures and the fact that open surgery is the gold standard in AA of ascending aorta twice. Please shorten these parts. Otherwise, you manuscript is well written. 

Author Response

Congratulations for your work. Although this study is an retrospective study from a national databasis, still it is quite interesting. You mention in your manuscript that similar studies from other national databases have already been published (eg. the Swedish one). However, I believe that your study adds knowledge to the existing. It is a well designed and presented study. I would omit some repetitions in the discussion. For example, you mention the advances of endovascular procedures and the fact that open surgery is the gold standard in AA of ascending aorta twice. Please shorten these parts. Otherwise, you manuscript is well written. 

Thank you very much for your hint. The paragraph was adapted as follows:

“Open surgical repair is still considered the gold standard for the treatment of pa-thologies involving the ascending aorta and aortic arch [[41,42]. However, a significant proportion of patients with these pathologies have relevant concomitant cardiopulmonary diseases that preclude these highly invasive procedures. The endovascular revolution began in the late 1980s and has become the gold standard for the treatment of descending aortic pathologies [43]. Likewise, the majority of patients with infrarenal aortic diseases is nowadays treated endovascularly in most Western countries. These successes have led to an extension of the endovascular concept to the more complex thoracoabdominal aortic segment. The first branched thoracic stent graft was performed in 1996 in a patient with an aortic dissection [44]. However, it is only in the last 10-15 years that this technology has become established, standardized, and widespread.

The aortic root and the ascending aorta constitute unique anatomical and phys-io-logical characteristics. The rapid blood flow, the high wall tension, the proximity to the aortic valve and coronary ostia, as well as the four-dimensional pulsatile motion during cardiac and respiratory cycles represent the ultimate challenges to the endovascular therapy [45]. Despite these challenges, attempts to treat the ascending aorta endovas-cularly have been made since 2000 and have gained increasing attention as an alternative treatment strategy, especially for critically ill patients [41]. In recent years, there have been growing case series on the endovascular treatment of non-dissected diseases of the as-cending aorta [42].The currently available prostheses in the market require specific an-atomical conditions. Moreover, most are not directly available off-the-shelf but are cus-tom-made for individual patients, posing a challenge in emergency situations due to the time required for production. Nevertheless, viable solutions exist with the appropriate equipment. For instance, in emergency situations where cerebral perfusion is assured (e.g., with veno-arterial ECMO), in situ laser fenestration can be used to address pa-thologies in the aortic arch [46].

With the endovascular aortic arch repair being established in selected patients, endovascular therapy has reached the ascending aorta and with it its last bastion. It is particularly interesting for patients with a high surgical risk and could offer these patients a potentially life-saving therapy that would otherwise not be surgically treated [13,43]. While the introduction of endovascular therapy in the descending aorta and abdominal aorta has led to a significant reduction in conventional open surgery, this trend has not been demonstrated for the ascending aorta or the aortic arch [4,7]. Likewise, there was no decreasing incidence for open repair of ascending or arch pathologies in Switzerland. It seems like the endovascular therapy in Switzerland is currently complementing open repair and does not lead to a decrease in open repair.

Diseases of the proximal aorta are complex and associated with high procedural risks. This requires tailor-made treatment strategies that are estabilshed by a multidisciplinary team, including cardiac surgeons, vascular surgeons, interventional radiologists, cardi-ologists, and anesthesiologists.”

Reviewer 3 Report

Comments and Suggestions for Authors

This paper examines the treatment transcription of ruptured thoracic aortic aneurysms in Switzerland. It deals with so-called big data, but the conclusions are similar to those of other countries and are not spectacular.

It is worthwhile as a record of the present situation of Switzerland.

Discuss how to treat the patient population with high van Walraven scores in light of these results. Specifically,

(1) Should we check van Walraven scores at medical examinations or other means and make an effort to pick up high-scoring patients?

(2) Whether should surgeons operate on patients with high van Walraven scores or not?

(3) It seems that the surgical mortality rate for nrAA cases is higher than in other countries. What do you think about this?

Comments on the Quality of English Language

Minor revision are to be considered.

Author Response

Reviewer 3.

This paper examines the treatment transcription of ruptured thoracic aortic aneurysms in Switzerland. It deals with so-called big data, but the conclusions are similar to those of other countries and are not spectacular.

It is worthwhile as a record of the present situation of Switzerland.

Thank you very much.

Discuss how to treat the patient population with high van Walraven scores in light of these results. Specifically,

(1) Should we check van Walraven scores at medical examinations or other means and make an effort to pick up high-scoring patients?

(2) Whether should surgeons operate on patients with high van Walraven scores or not?

Answer to (1) and (2): As the Walraven score is calculated from administrative data (see Reference), which is only collected during the course of patients treatment, it will be difficult to use it in everyday clinical practice before a planned operation. It can be used for post-discharge analysis within the framework of studies and  and serves the better comparability between international study but is not intended for to use as a prognostic marker. 

Reference: van Walraven, C.; Austin, P.C.; Jennings, A.; Quan, H.; Forster, A.J. A Modification of the Elixhauser Comorbidity Measures into a Point System for Hospital Death Using Administrative Data. Med. Care 2009, 47, 626–633, doi:10.1097/MLR.0b013e31819432e5.

(3) It seems that the surgical mortality rate for nrAA cases is higher than in other countries. What do you think about this?

We are unable to verify the reviewers statement. With an overall mortality rate of 2% rather the opposite is the case, as can be seen from the national American or Dutch figures, for example: references:

  1. Beyer SE, Secemsky EA, Khabbaz K, Carroll BJ. Elective ascending aortic aneurysm repair outcomes in a nationwide US cohort. Heart. 2023 Jun 26;109(14):1080-1087. doi: 10.1136/heartjnl-2022-322033. PMID: 36928243.
  2. Gökalp AL, Thijssen CGE, Bekkers JA, Roos-Hesselink JW, Bogers AJJC, Geuzebroek GSC, Houterman S, Takkenberg JJM, Mokhles MM. Male-female differences in contemporary elective ascending aortic surgery: insights from the Netherlands Heart Registration. Ann Cardiothorac Surg. 2023 Nov 27;12(6):577-587. doi: 10.21037/acs-2022-adw-fs-0139. Epub 2023 Jun 27. PMID: 38090337; PMCID: PMC10711412.

We have added this to the discussion and references accordingly:

In our cohort, an overall mortality rate of 2.0% was observed in surgically treated nrAAs and is therefore lower than in other countries such as the United States or the Netherlands [24,25]. (lines: 247ff).

Reviewer 4 Report

Comments and Suggestions for Authors

The article is well-written and organized in a clear and reader-friendly way.

The main suggestion that I have to report is that a brief chapter about the main techniques of open and endovascular treatment of ascending and aortic arch aneurysms will make the manuscript more complete.

I will suggest to take into account different predictive factors, maybe in smaller cohorts, or to try different statistical methods to find more relevant predictive factors or original correlations (for example the better outcome in smaller hospitals vs university hospitals is interesting).

Comments on the Quality of English Language

The use of English is well-done and the article is clear.

I detected only a few mistakes:

Line 137: 73.6% of patients were male

Line 148-9: the same

Author Response

Reviewer 4.

The article is well-written and organized in a clear and reader-friendly way.

The main suggestion that I have to report is that a brief chapter about the main techniques of open and endovascular treatment of ascending and aortic arch aneurysms will make the manuscript more complete.

Thanks a lot for this constructive comment. We appreciate your proposal. We added the following paragraph to the introduction:  

“Of the various techniques, the best known are the Bentall- and the David procedure. In the Bentall procedure, the aortic valve, the aortic root and the ascending aorta are replaced with a composite graft [9]. In the David procedure, the enlarged aortic root is replaced with a Dacron graft, leaving the natural aortic valve intact [10,11]. In addition to open surgical repair, endovascular treatments are also available in which a stent-graft is used to seal the aneurysm and reinforce the aortic wall, also known as thoracic endovascular aortic repair (TEVAR) [7,12]. Endovascular procedures are minimally invasive and are based on ad-equate sealing in healthy aortic segments. Insufficient sealing can lead to incomplete exclusion of the aneurysm. Sealing is particularly challenging in the aortic sinus, where the coronary artery originates, and in the aortic arch, where the supra-aortic branches originate. The choice of treatment method depends on several factors, including the patient's condition, the anatomy of the aneurysm as well as the treating physician's preference.”

I will suggest to take into account different predictive factors, maybe in smaller cohorts, or to try different statistical methods to find more relevant predictive factors or original correlations (for example the better outcome in smaller hospitals vs university hospitals is interesting).

Thank you for this excellent comment. It is indeed a good idea to conduct some alternative analyses to get better insights into the data. As proposed by reviewer #1 we conducted a multilevel multivariable logistic regression analysis with sex as a level. Details described above.

Further, it might be plausible, that patients treated at major hospitals were different in ways not reflected by the data and that there were dependencies between these variables that do not justify hospital level as a factor variable into the analysis. Therefore, we conducted another analysis with hospital level as a level variable in a multilevel multivariable analysis. The results can be seen here:

Again, the results are very similar to the originally presented model and we see no reason to treat the data differently. We believe that the chosen model is appropriate and represents the data most appropriately.

Comments on the Quality of English Language

The use of English is well-done and the article is clear.

I detected only a few mistakes:

Line 137: 73.6% of patients were male

Line 148-9: the same

Thank you for spotting these typos that now have been corrected.
